# Plant-Derived Oleanolic Acid (OA) Ameliorates Risk Factors of Cardiovascular Diseases in a Diet-Induced Pre-Diabetic Rat Model: Effects on Selected Cardiovascular Risk Factors

**DOI:** 10.3390/molecules24020340

**Published:** 2019-01-18

**Authors:** Mlindeli Gamede, Lindokuhle Mabuza, Phikelelani Ngubane, Andile Khathi

**Affiliations:** Department of Human Physiology, School of Laboratory Medicine and Medical Sciences, College of Health Sciences, University of KwaZulu-Natal, Durban X54001, South Africa; 213571877@stu.ukzn.ac.za (M.G.); 211509843@stu.ukzn.ac.za (L.M.); Ngubanep1@ukzn.ac.za (P.N.)

**Keywords:** pre-diabetes, dyslipidemia, cardiovascular disease, arterial pressure and inflammation

## Abstract

The pathogenesis of prediabetes is associated with risk factors such as chronic consumption of an unhealthy diet. Recent studies have reported that diet-induced pre-diabetes is also associated with risk factors of cardiovascular complications, hence this study was aimed at evaluating the effects of oleanolic acid (OA) on pre-diabetes rats. Pre-diabetes was induced by chronic exposure of Sprague Dawley rats (SD) to high-fat high-carbohydrate diet (20 weeks), whereas the non-pre-diabetes control (NC) was given standard rat chow. Pre-diabetes animals were grouped into five groups namely prediabetes control (PC), metformin treated (Met), metformin with diet intervention (Met + DI), oleanolic acid treated (OA), and oleanolic acid with diet intervention (OA + DI) then treated for 12 weeks. At the end of treatment, all animals were sacrificed where organs and tissues were harvested for biochemical analysis and histological studies. The results showed that PC had a significantly higher triglycerides (TGs), low density lipoprotein cholesterol (LDL-C, interleukin-6(IL-6), tumor necrosis factor alpha (TNFα), C-reactive protein (CRP), mean arterial pressure (MAP) and hearts weights in comparison to NC (*p* < 0.05). However, the administration of OA, in both the presence and absence of dietary intervention showed a significant decrease in TGs, LDL-C, IL-6, TNFα, CRP, MAP, hearts weights (*p* < 0.05). In conclusion, the administration of OA was able to lower the risks of developing CVDs in pre-diabetes rat model through ameliorating dyslipidaemia, oxidative stress, hypertension, and low-grade inflammation. Therefore OA has the potential to be used as an alternative treatment to prevent the onset of CVDs during pre-diabetes stage even in the absence of dietary and lifestyle intervention.

## 1. Introduction

Cardiovascular diseases (CVDs) are among the leading causes of mortality and morbidity in type 2 diabetes patients worldwide [1]. Approximately more than half of type 2 diabetic patients suffer from cardiovascular diseases such as coronary heart disease, cardiac hypertrophy, atherosclerosis and hypertension [2,3]. These complications are traditionally associated with overt type 2 diabetes mellitus (T2DM), however recent studies have shown that pre-diabetes is also associated with augmented risk factors of developing CVDs. The CVD risk factors that are associated with pre-diabetes include insulin resistance, obesity, dyslipidaemia and oxidative stress [4]. Furthermore, insulin resistance is also directly linked with the development of cardiovascular complications such as increased blood pressure as a consequence of nitric oxide synthase inhibition and decreased nitric oxide (NO) bioavailability [5,6]. Previous studies have shown that elevation of blood glucose may lead to the generation of reactive oxygen species (ROS) which is linked with vascular endothelial dysfunction and subsequently the decrease in NO bioavailability [7]. This is associated with cardiovascular complications such as hypertension and subsequent cardiac hypertrophy [8]. In addition, insulin resistance is also associated with dyslipidaemia which is characterized by increased plasma triglycerides, increased low-density lipoprotein cholesterol and decreased high-density lipoproteins cholesterol [9,10]. The decrease in HDL-C correlates with an increase in LDL-C which is traditionally associated with the pathogenesis of CVDs such as atherosclerosis accompanied by systemic inflammation [11]. Low-grade inflammation including augmented cardiac tissue C-reactive protein, tumor necrosis factor alpha, and interlukin-6 is also associated with the pathogenesis of macrovascular complications such as peripheral resistance, high blood pressure and subsequent cardiac hypertrophy [12]. Traditional treatments for diabetes related cardiovascular complications include the combination of dietary modification as well as a cocktail of pharmacological agents that can include insulin sensitizers, beta blockers, angiotensin converting enzyme (ACE) blockers and statins [13,14]. The efficacy of these drugs is often impeded by low patient compliance resulting from neglecting dietary modification as well as the consumption of multiple medications at a single time [15]. There is, therefore, a need for diabetes drugs that can remain therapeutic in the absence of dietary modification as well as also prevent the onset of cardiovascular complications. Previous studies in our laboratory have indicated *Syzygium aromaticum*-derived oleanolic acid ameliorated insulin resistance and restored glucose homeostasis in a diet-induced pre-diabetic rat model [16,17]. However, the effects of OA on cardiovascular disease risk factors in pre-diabetic animals remain unknown. Therefore, this present study was designed to evaluate the therapeutic effects of plant-derived OA on diabetes-related cardiovascular complications in an HFHC diet-induced pre-diabetic rat model, through monitoring the effect of OA on selected risk factors of CVDS.

## 2. Results

### 2.1. Fasting Blood Glucose

Fasting blood glucose of all experimental groups was measured at the start of the treatment period week 0 and at the end (week 12) prior to termination of the study. At the beginning of the treatment period the pre-diabetic groups had a significant higher fasting blood glucose in comparison to the normal control (NC) (*p* < 0.05) (see Figure 1A).

During week 12, the PC group had a significantly higher fasting blood glucose to NC (PC vs NC) (*p* < 0.05). However, OA treated groups (OA and OA + DI) showed a significant decrease in fasting blood glucose when compared to PC (*p* < 0.05) (see Figure 1B). The effects of OA and OA + DI were found to be comparable with those of MET and MET + DI.

### 2.2. Mean Arterial Pressure

Mean arterial pressure (MAP) of all experimental groups was monitored at the start of the treatment period, (week 0) and at the end of the treatment period (12 weeks). At the beginning of the treatment period, all pre-diabetic groups had a significant higher MAP in comparison to normal control (NC) (see Figure 2A). However, at the end of the treatment period (week 12), only the PC group had a significantly higher mean arterial pressure in comparison to NC (PC vs NC) (*p* < 0.05). The OA-treated groups (OA and OA + DI) showed a significant decrease in mean arterial pressure when compared to PC (*p* < 0.05) (see Figure 2B). The same effects were observed for MET and MET + DI.

### 2.3. Heart and Body Weights; Cardiac C-Reactive Protein (CRP)

At the end of the treatment period, animals from all experimental groups were sacrificed, and hearts were weighed for the calculation of heart to body weight ratio. The results showed that the PC had significantly higher heart weights when compared to NC (*p* < 0.05). In contrast, heart/body weight ratios of PC were significantly lower when compared to NC (*p* < 0.05). However, OA-treated groups (OA and OA + DI) had a significantly lower heart weights and inversely higher heart/body ratios in comparison to PC (*p* < 0.05).

Cardiac tissue C-reactive protein (CRP) concentration of all the experimental groups was measured at the end of the treatment period (week 12). The results showed that the CRP concentration of PC was significantly higher than of NC (*p* < 0.05). In all groups where OA was administered (OA and OA + DI), there was a significant decrease in CRP concentration when compared to PC (*p* < 0.05). Interestingly, the administration of MET and MET + DI did not significantly reduce CRP levels (see Table 1).

### 2.4. Plasma Lipid Profile

At the end of the treatment period (week 12), plasma lipids were analyzed in all experimental groups. The results showed that the PC had significantly higher bad cholesterol such as LDL-C and TGs when compared to NC (*p* < 0.05). However, PC had significant decreased good cholesterol such as HDL-C in comparison to NC. The OA treated group (OA and AO + DI) showed a significant decrease in LDL-C and TGs when compared to PC (*p* < 0.05). In addition, OA also significantly improved HDL-C (*p* < 0.05) to within range of the NC group. Additionally, while the administration of MET alone did not significantly reduce LDL levels, the administration of MET + DI significantly reduced plasma LDL (*p* < 0.05) (see Table 2).

### 2.5. Inflammatory Markers (IL-6 and TNFα)

All experimental groups were sacrificed at the end of the treatment period (week 12) and the plasma IL-6 and TNFα concentration was measured. The results showed that both plasma IL-6 and TNFα concentration of PC was significantly higher than of NC (*p* < 0.05). 

In all groups where OA was administered (OA and OA + DI), there was a significant decrease in plasma IL-6 and TNFα concentration when compared to PC OA treated group had no significant difference when compared to PC (see Figure 3A). However, the administration of OA + DI showed a significant decrease in plasma TNF concentration when compared to PC (*p* < 0.05) (see Figure 3B). Furthermore, the administration of MET did not significantly reduce TNF levels, the administration of MET + DI significantly reduced plasma TNF levels by comparison to PC (*p* < 0.05). 

### 2.6. Oxidative Stress

At the end of the treatment period, all experimental groups were sacrificed for heart MDA, SOD and GPx measurements. The results showed that the PC had a significantly higher MDA concentration when compared to NC (*p* < 0.05). Consequently, PC had a significantly lower SOD and GPx concentrations when compared to NC (*p* < 0.05). However, the administration of OA had a significantly decreased MDA concentration when compared to PC (*p* < 0.05) and subsequently the improvement of antioxidant status was observed by significant increase SOD and GPx in both OA treated groups (OA and OA + DI) which were comparable with metformin treated groups (Met and Met + DI) (Figure 4).

## 3. Discussion

The present study investigated the effects of *Syzygium aromaticum*-derived OA on selected cardiovascular risk factors in a HFHC diet-induced pre-diabetic rat model. Prolonged exposure to unhealthy diets such as those high in carbohydrates and saturated fats have been implicated in the pathogenesis of pre-diabetes which later develops to overt T2DM if left untreated. Studies have shown that the onset of prediabetes is accompanied by the of cardiovascular complications such as hypertension and atherosclerosis [18]. This suggests that predisposing risk factors of CVDs including dyslipidaemia, oxidative stress, low-grade inflammation and endothelial dysfunction arise prior to the diagnosis of T2DM [19]. Prolonged sustained hyperglycaemia has been traditionally considered as the source of all diabetic complications [20]. However, the consumption of HFHC diet has been previously reported to induce metabolic syndrome with cardiovascular complications such as high blood pressure and systemic inflammation [21]. Moreover, in our laboratory a rat model of pre-diabetes induced by chronic consumption of a HFHC diet has been reported to possess similar symptoms [22]. This suggests that exposure to HFHC diet is also a risk factor in the development of CVDs. A combination of diet intervention with pharmacological treatment has been shown to be effective in managing pre-diabetes and attenuating the associated cardiovascular risk factors [23]. Pharmacotherapy such as the use of medicinal plant bioactive compounds including triterpenes has been widely used in diabetes mellitus research [24]. According to previous studies, triterpenes possess anti-diabetic properties, however, there are limited clinical studies that can further elucidate these properties [25]. Triterpenes such as urosolic acid (UA), maslinic acid (MA) and oleanolic acid(OA) have been reported to possess anti-diabetic properties in STZ-induced diabetes [26]. More recently, OA has been found to ameliorate insulin resistance in a diet–induced prediabetic rat model [17]. However, the effects of OA on CVD risk factors such as oxidative stress, dyslipidaemia, and low-grade inflammation have not yet been investigated in a pre-diabetes rat model. Hence the current study evaluated the therapeutic effects of OA on selected cardiovascular disease risk factors in a diet-induced pre-diabetes rat model. The pathogenesis of T2DM-related complications is traditionally attributed to dysregulation of glucose homeostasis which is attributed to insulin resistance and the shortage of insulin production at a later stage [27]. In this study, we observed that pre-diabetic animals had impaired fasting blood glucose. This may be attributed to the prolonged exposure of animals to HFHC diet which has been previously reported to cause the ectopic lipid accumulation in organs including liver and skeletal muscle [28]. The ectopic lipid accumulation is associated with impairment of insulin signaling pathway and subsequently the decrease in muscle glucose uptake which is implicated in the elevation of fasting blood glucose with hyperinsulinemia [29]. However, the administration of OA resulted on normalization of fasting blood glucose levels. This may be attributed to the previously reported anti-hyperglycaemic properties of triterpenes which includes myocytes and adipocytes glucose uptake through promoting glucose transpoter-4(GLUT-4) translocation to the cellular membrane [30]. Moreover, triterpenes are also known to inhibit the protein tyrosine phosphatase enzymes [31]. These enzymes are involved in the dephosphorylation of the beta-subunit of the insulin receptor and subsequently result in the impairment of the insulin dependent glucose uptake [32]. Therefore, the inhibition of these enzymes may result in normalization of plasma glucose without hypoglycaemia. The hyperglycaemic conditions in pre-diabetes are associated with the generation of oxidative stress and abnormalities in lipid metabolism which is directly linked with the pathogenesis of CVDs [21]. Oxidative stress is a condition where the production of reactive oxygen species (ROS) overcome the endogenous antioxidants in cells and tissues [33]. During pre-diabetes, hyperglycaemic conditions favor the over production of ROS through mitochondrial glucose oxidation [34]. In this study malonaldehyde (MDA) which is a biproduct of lipid peroxidation was used to quantify the presence of ROS [34]. Interestingly we observed that pre-diabetic animals had significant high concentrations of MDA. This correlated with the suppression of endogenous antioxidants including SOD and GPx. However, the administration of OA led to the reduction of MDA and improvement of the antioxidant status. These results suggest that OA may be exhibiting antioxidant activity through scavenging hydroxyl by donating hydrogen molecule [35]. This may explain the observed improvement in the antioxidant profile in OA-treated animals. The pre-diabetic state is also characterized by hyperglycaemic conditions that co-exist with ectopic intramyocellular lipid accumulation and increase in cellular diacyl glycerol formation which leads to activation of protein kinase-theta [36]. This is associated with the inhibition of skeletal muscle cells insulin signaling and subsequently the development of muscle insulin resistance [36,37]. Muscle insulin resistance is directly linked with increased hepatic *de novo*-lipogenesis and subsequently the increase in circulating triglycerides, very low-density lipoprotein-cholesterol, with a decrease in high density lipoprotein-cholesterol [38]. The current study also found that pre-diabetic animals had derangements in lipid metabolism including high levels of circulating triglycerides, low density lipoprotein-cholesterol and decreased high density lipoprotein-cholesterol. This may suggest that diet-induced pre-diabetes are also led to the derangement in lipid metabolism and subsequently abnormalities in cholesterol profile which are known to be pro-atherogenic [39,40]. HDL-C plays an important role in the prevention of the onset of cardiovascular disease through transporting LDL-C from cardiomyocytes and blood vessels to the liver where it is cleared [35]. Therefore, the reduction of HDL-C increases the risks of developing cardiovascular diseases such as atherosclerosis due to accumulation of cholesterol in blood vessels [35,36]. However, the administration of OA ameliorated lipid metabolism possibly through lowering circulating triglycerides and LDL-C as well as normalizing HDL-C. This may be attributed to the previously reported ability of OA to suppress CCAAT/enhancer binding proteins-α (C/EBPα) and peroxisome proliferator activated receptor-γ (PPARγ) expression which are known to be pro-lipogenesis in animal models of metabolic syndrome [41]. The downregulation of C/EBαP and PPARγ have been reported to decrease in hepatic lipid synthesis while improving HDL-C fuction, hence lowers the CVDs risks [42]. 

Dysregulation of lipid metabolism accompanied with oxidative stress is implicated in the development of CVDs such as endothelial dysfunction and high blood pressure [43]. In the present study, exposure of animals to HFHC diet resulted in an increase of the mean MAP. This increase in MAP may be attributed to atherogenic dyslipidaemia which leads to the build-up of plaque in arterial walls [44]. However, the administration of OA resulted in decreased MAP. These observations may be attributed to OA acting as a vasodilator through enhancing the expression of nitric oxide synthase enzyme (eNOS) and subsequently the production of NO by the arterial endothelial lining [45]. Interestingly, we found that these findings were comparable with the results of the previous studies which also reported that plant-derived triterpenoids possess antihypertensive properties on the STZ-induced diabetic rat model [46]. In addition, triterpenoids have also been reported to increase the production of NO through increasing the expression of nitric oxide synthase [47].

Arterial wall plaque accumulation may lead to increased peripheral resistance which is associated with cardiac work overload and subsequent cardiac hypertrophy which is characterized by thickening of the ventricles as well as increased heart sizes [48]. In this current study we observed that exposure of rats to HFHC diet led to increase in body weight and heart sizes. The increase in heart size during pre-diabetes may be attributed to compensatory hyperinsulinemia and augmented TNFα which transform insulin growth factor-1(IGF-1) and lead to hyperplasia of cardiomyocytes [48,49]. However, the administration of OA also resulted in both decrease in heart weights and reduction of body weight. The current findings are comparable with the results from a previous study which reported that OA activates insulin receptor-1 (IR-1) which lowers circulating insulin and IGF-1 levels and subsequently reduces cardiac hypertrophy in animal models of obesity and metabolic syndrome [50]. 

The co-existence of low grade inflammation and insulin resistance is strongly associated with high risks of developing cardiovascular complications [5,51]. Augmented circulatory interleukins and growth factors such as IL-6 and TNFα respectively have been reported in obesity [52]. A previous study done in our laboratory showed that the consumption of an HFHC diet leads to glucose intolerance and obesity which is directly linked with ectopic deposition of fats [17]. Ectopic visceral fats are infiltrated by macrophages and subsequently high production of pro-inflammatory cytokines including IL-6, and TNFα which further exacerbate inflammation [53]. In addition, these pro-inflammatory cytokines are implicated in increased cardiac tissue CRP which is strongly associated with the onset of cardiovascular diseases [54]. However, the administration of OA normalized plasma concentrations of these inflammatory cytokines and cardiac CRP. These results may be attributed to the anti-inflammatory properties of triterpenoid through normalizing the recruitment of immune cell in to the circulation there by inhibiting the activation and secretion of pro-inflammatory cytokines in T-cells as well as macrophages [47]. 

## 4. Materials and Methods

### 4.1. Drugs and Chemicals

All chemicals and reagents were sourced from standard pharmaceutical suppliers and were of analytical grade, including isofor (Safeline Pharmaceuticals (Pty) Ltd., Roodeport, South Africa), liquid nitrogen (Chemistry Department UKZN, Wstville, South Africa), ELISA kits (Elabscience, Mzansi Medical Laboratories, South Africa), metformin, phosphoric acid (Sigma, Durban, South Africa) thiobarbituric acid(Sigma, Durban, South Africa), hydroloric acid (Sigma, Durban, South Africa), and butanol (Sigma, Durban, South Africa). 

### 4.2. Extraction Method

OA was extracted from *Syzygium aromaticum* [(Linnaeus) Merrill & Perry] [Myrcene] (cloves) using an established protocol from Khathi et al. [55]. Briefly, air-dried *Syzygium aromaticum* flower buds (500 g) were milled and sequentially extracted twice at 24 h intervals at room temperature using 1 L dichloromethane (DCM), and ethyl acetate (720 mL) on each occasion. Subsequently, the extract was concentrated under reduced pressure at 55 ± 1 °C using a rotary evaporator (RE20, Labotech (PTY) LTD, Johannesburg, South Africa,) to yield dichloromethane solubles (DCMS) and ethyl acetate solubles (EAS). The EAS containing mixtures of oleanolic/ursolic acid and methyl maslinate/methyl corosolate was purified by silica gel 60 column chromatography with hexane: ethyl acetate solvent systems of 7:3. This yielded OA which was further purified by recrystallization from chloroform-methanol (1:1, *v*/*v*). The structure of OA was confirmed by spectroscopic analysis using ^1^D and ^2^D, ^1^H and ^13^C nuclear magnetic resonance (NMR) spectroscopic experiments. OA that had a purity of greater than 95% was used in the study.

### 4.3. Animal Studies

Male Sprague-Dawley rats (130–160 g) (5–6 weeks old) (*n* = 36) used in this study were bred and housed in the Biomedical Research Unit of the University of KwaZulu-Natal. The animals were maintained under standard laboratory conditions of constant temperature (22 ± 2 °C), CO_2_ content (<5000 p.m.), relative humidity (55 ± 5%) and illumination (12 h light/dark cycle, lights on at 06h00 and switched off at 18h00). The noise level was maintained at less than 65 Db. The animals were allowed access to food and fluids *ad libitum*. All animal procedures and housing conditions were approved by the Animal Research Ethics Committee of the University of KwaZulu-Natal (ethics no: AREC/035/016M). The animals could acclimatize to their new environment for 1 week before the commencement of the trial.

#### 4.3.1. Experimental Design

At the end of 20 weeks HFHC diet induced pre-diabetic male Sprague-Dawley rats were divided into six groups (*n* = 6): pre-diabetic control group (PC), diet intervention group (DI), metformin treated group (Met), metformin and diet intervention group (Met + DI), OA treated group (OA) as well as OA and diet intervention group (OA + DI). The animals that were on standard chow from the beginning of the study were considered as non-pre-diabetes control group (NC) (*n* = 6) (see Appendix A).

#### 4.3.2. Experimental Protocol

The experimental period was 12 weeks and began at the end of the 20-week pre-diabetes induction period. Animals were treated every third day. Every fourth week from the beginning of the experimental period, parameters including fasting blood glucose, total cholesterol and triglyceride concentrations were measured in all groups. The systolic and diastolic pressure of all experimental animals was measured using non-invasive tail cuff method and the mean arterial pressure was calculated using the formula:MAP = [(2 × diastolic) + systolic]/3(1)

#### 4.3.3. Experimental Procedures

##### Blood Collection and Tissue Harvesting

For blood collection, all animals were anaesthetized with Isofor (100 mg/kg) via a gas anaesthetic chamber (Biomedical Resource Unit, UKZN, Durban, South Africa) for 3 min. Blood was collected by cardiac puncture and then injected into individual pre-cooled heparinized containers. The blood was then centrifuged (Eppendorf Centrifuge 5403, Hamburg, Germany) at 4 °C, 503 g for 15 min. Plasma was collected and stored at −80 °C in a Bio Ultra freezer (Snijers Scientific, Tilburg, The Netherlands) until ready for biochemical analysis. Thereafter, the heart was removed, rinsed with cold normal saline solution and snap frozen in liquid nitrogen before storage in a Bio Ultra freezer (Snijers Scientific) at −80 °C until further biochemical analysis.

### 4.4. Biochemical Analysis

SOD, GPx, TNFα, IL-6, and C-reactive protein were analysed using their respective rat ELISA kits (Elabscience Biotechnology Co., Ltd., Texas, United State of America (USA) according to the manufacturer’s instructions. The plasma lipid profile was determined through the measurement of triglycerides (TGs), total cholesterol [56], high density lipoproteins cholesterol (HDL-C) and low-density lipoproteins cholesterol (LDL-C) at Global clinical and viral laboratories (Amanzimtoti, Durban, South Africa) for quantification. The fasting blood glucose was measured by tail pick method using OneTouch select glucometer.

### 4.5. MDA Assay

Heart tissues (50 mg) were homogenized in 500 mL of 0.2% phosphoric acid. The homogenate was centrifuged at 400 g for 10 min. Thereafter, 400 mL of the homogenate was supplemented with 400 mL 2% phosphoric acid and then separated into two glass tubes, each receiving equal volumes of the solution. Subsequently, 200 mL of 7% phosphoric acid was added into both glass tubes followed by the addition of 400 mL of thiobarbituric acid (TBA)/butylated hydroxytoluene (BHT) into one glass tube (sample test) and 400 mL of 3 mM hydrochloric acid (HCl) into the second glass tube (blank). To ensure an acidic pH of 1.5, 200 mL of 1M HCl was added to sample and blank test tubes. Both solutions were heated at 100 °C for 15 min, and allowed to cool to room temperature. Butanol (1.5 mL) was added to the cooled solution; the sample was vortexed for 1 min to ensure rigorous mixing and allowed to settle until two phases could be distinguished. The butanol phase (top layer) was transferred to Eppendorf tubes and centrifuged at 13,200 g for 6 min. The samples were aliquoted into a 96-well microtiter plate in triplicate and the absorbance was read at 532 nm (reference 600 nm) on a BioTek mQuant spectrophotometre (Biotek, Johannesburg, South Africa). The absorbance from these wavelengths was used to calculate the concentration of MDA using Beer’s Law:Concentration of MDA = Average Absorbance Absorption/coefficient (156 nm)(2)

### 4.6. Statistical Analysis

All data was expressed as means ± S.E.M. Statistical comparisons were performed with GraphPadInStat Software (version 5.00, Graph Pad Software, Inc., San Diego, CA, USA) using One-way analysis of variance (ANOVA) followed by Bonferroni *post hoc* comparison test. A value of *p* < 0.05 was considered statistically significant.

## 5. Conclusions

Taken together, the findings of this study further confirm that plant-derived OA possesses beneficial cardiovascular effects through the restoration of glucose homeostasis. This study also showed that the administration of OA can ameliorate cardiovascular function through mechanisms including regulation of MAP in the pre-diabetic state and thus prevent the onset of diabetic cardiovascular complications. Furthermore, this study elucidated that OA can suppress low-grade inflammatory mediators and improve lipid metabolism. The results of this study warrant further investigations into the use of OA in the pre-diabetic state to prevent the onset of diabetes-related cardiovascular complications.

## Figures and Tables

**Figure 1 molecules-24-00340-f001:**
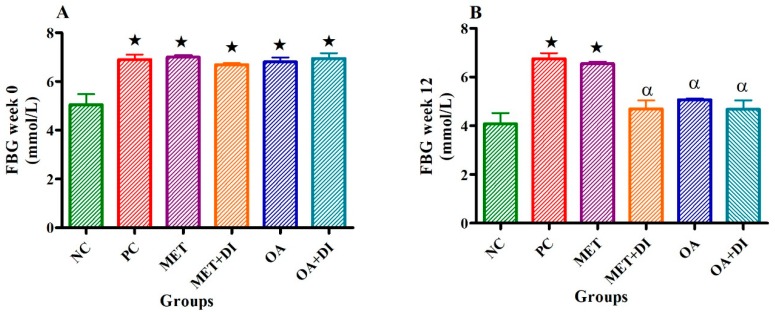
Effects of OA and Met (*n* = 6, per group) on fasting blood glucose of rats that continued with HFHC diet and those that changed diet or had diet intervention at week 0 (**A**) and week 12 (**B**) respectively. Values are presented as standard error of mean ± SEM. 
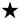
 = *p* <0.05 denotes comparison with NC; α = *p* < 0.05 denotes comparison with PC.

**Figure 2 molecules-24-00340-f002:**
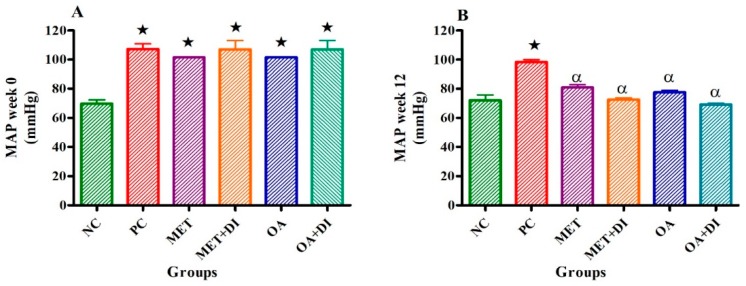
Effects of OA and Met (*n* = 6, per group) on mean arterial pressure of rats that continued with HFHC diet and those that changed diet or had diet intervention at week 0 (**A**) and week 12 (**B**) respectively. Values are presented as standard error of mean ± SEM. 
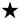
 = *p* < 0.05 denotes comparison with NC; α = *p* < 0.05 denotes comparison with PC.

**Figure 3 molecules-24-00340-f003:**
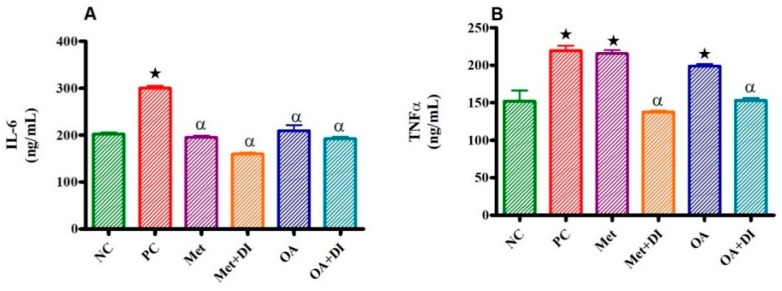
Effects of OA and Met (*n* = 6, per group) on IL-6 (**A**) and TNFα (**B**) concentrations of rats that continued with HFHC diet and those that changed diet or had diet intervention respectively Values are presented as standard error of mean ± SEM. 
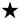

*p* < 0.05 denotes comparison with NC; α *p* < 0.05 denotes comparison with PC.

**Figure 4 molecules-24-00340-f004:**
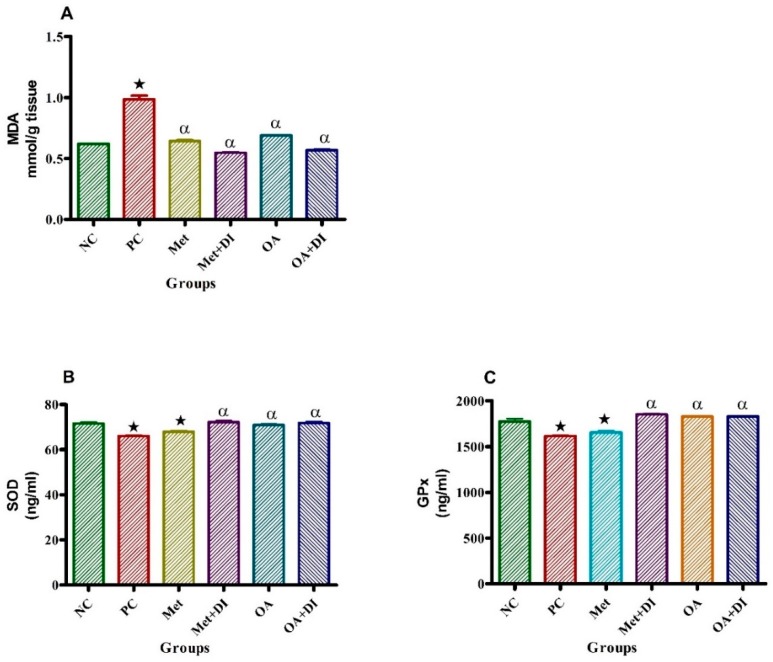
Effects of OA and Met (*n* = 6, per group) on MDA, SOD and GPx of rats that changed from HFHC diet to normal diet. Values are presented as standard error of mean ± SEM. Non-pre-diabetes control (NC), pre-diabetes control (PC), metformin treated with dietary intervention (Met + DI) and oleanolic acid treated with dietary intervention (OA + DI). 
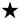

*p* < 0.05 denotes comparison with NC; α *p* < 0.05 denotes comparison with PC.

**Table 1 molecules-24-00340-t001:** Effects of OA and Met (*n* = 6, per group) on heart sizes, body weights and cardiac CRP of rats that continued with HFHC diet. Values are presented as standard error of mean ± SEM. Non-pre-diabetes control (NC), pre-diabetes control (PC), metformin treated (Met), Metformin treated with dietary intervention (Met + DI), oleanolic acid treated (OA) and oleanolic acid treated with dietary intervention (OA + DI).

Experimental Group	Heart Weights (g)	Body Weights (g)	Heart/Body Ratio (H/B)	Cardiac CRP (ng/mL)
**NC**	1.56 ± 0.08	387.50 ± 11.18	0.0040 ± 0.00029	9.63 ± 1.08
**PC**	1.72 ± 0.05 *	679.75 ± 78.52 *	0.0027 ± 0.00029 *	17.24 ± 0.35 *
**Met**	1.62 ± 0.03 α	500.50 ± 2.59 *α	0.0032 ± 0.00028 α	16.61 ± 0.17 *
**Met + DI**	1.73 ± 0.01 *	443.00 ± 13.86 *α	0.0039 ± 0.00012 α	15.03 ± 0.27 *
**OA**	1.61 ± 0.12 α	516.75 ± 8.28 *α	0.0031 ± 0.00019 α	12.10 ± 0.45 α
**OA+DI**	1.49 ± 0.04 α	434.25 ± 12.19 α	0.0034 ± 0.00011 α	9.56 ± 0.58 α

* *p* < 0.05 denotes comparison with NC; α *p* < 0.05 denotes comparison with PC.

**Table 2 molecules-24-00340-t002:** Effects of OA and Met on the lipid profile of rats that continued with HFHC diet. Values are presented as standard error of mean ± SEM. Non-pre-diabetes control (NC), pre-diabetes control (PC), metformin treated (Met), metformin treated with dietary intervention (Met + DI), oleanolic acid treated (OA) and oleanolic acid treated with dietary intervention (OA + DI).

Experimental Groups (*n* = 6)	TGs (mmol/L)	Total-C (mmol/L)	HDL-C (mmol/L)	LDL-C (mmol/L)
NC	1.23 ± 0.03	4.15 ± 0.05	1.71 ± 0.12	2.39 ± 0.03
PC	5.62 ± 0.32 *	4.12 ± 0.08	0.85 ± 0.04 *	8.88 ± 0.19 *
Met	3.29 ± 0.19 *α	4.12 ± 0.03	0.97 ± 0.04	5.93 ± 0.04 *α
Met + DI	0.91 ± 0.04 *α	4.10 ± 0.01	1.23 ± 0.04 α	1.89 ± 0.05α
OA	1.94 ± 0.15 α	4.05 ± 0.06	1.88 ± 0.02 α	5.69 ± 0.07 *
OA+DI	1.08 ± 0.05 α	4.10 ± 0.04	1.76 ± 0.02 α	2.04 ± 0.05 α

* *p* < 0.05 denotes comparison with NC; α *p* < 0.05 denotes comparison with PC.

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
