# Peer review of "Plant-Derived Oleanolic Acid (OA) Ameliorates Risk Factors of Cardiovascular Diseases in a Diet-Induced Pre-Diabetic Rat Model: Effects on Selected Cardiovascular Risk Factors"

_molecules, 2019, doi:10.3390/molecules24020340_

Round 1
Reviewer 1 Report
Comments to article: Plant-derived oleanoic acid (OA) ameliorates risk factors of cardiovascular diseases in a diet-induced pre-diabetic rat model: effects on selected cardiovascular risk factors
Dear Authors,
I would like to congratulate you on the fair attempt to accurately describe the findings of your study. Please refer to my comments/suggestions below. I hope it helps.
Major comment;
Overall: This manuscript requires excessive editing (symbols, abbreviations, probability values) and proof reading in English before proceeding any further.
1. I fail to see how is this manuscript significantly different (not form the statistical perspective) from the previous two publications acknowledged and referenced by authors (references 16 and 17)? Please provide an updated clarification in the introduction section of this manuscript.
2. I suggest that authors include the table of comparisons of all values that can be compared before and after the treatment within the groups. I would suggest that authors consider using the paired Students’ t-test to observe the changes that can occur over the period of time rather than just looking at the differences between the groups. It might be also useful to look a the percentage of change within the group and perform the appropriate statistical analysis.
3. I suggest that authors also consider performing some relationships analysis between the observed biomarkers.
Abstract requires revising and adding the actual results and discussion/conclusion section. It seems that only surface background information on introduction and methods section is provided. Please revise appropriately so it reflects the actual findings of the manuscript.
Introduction
I found the information in this section very interesting. Nevertheless, it requires re-write as it only provides the information about the T2DM but the information on the Oleanolic acid and relationship to the CVD. Although authors have stated that their previous studies have indicated that there are beneficial aspects of OA, there is no information on the previous findings.
Results
Please insert p values for all non-significant observations and highlight (with *) all significant observations
Discussion
In this section, I suggest that authors make a clear distinction between the animal models and human trials as the information presented here is very convoluted and does not provide clear description of either findings. One of the ways to do this is to use the animal trials as n indication of potential mechanisms of action while human observations as a cause and effect.
Methods
280 – According to the statement (please see minor comments section comment about acclimatisation) it appears that the NC rats were than put on the standard rats chow (which is usually LF) but this is only after the HFHC treatment initially. If this is correct, then the NC group is not really representing the control group? Please clarify and elaborate more.
Minor comments;
Results
67-69 Please reword this paragraph to state that plasma glucose was higher in the PC group than the other rather than increase. The way it is stated, results should be included at the start of the study as well which is not the case.
77-81 Although there was a higher MAP between the groups, was there comparison made between the start and the end?
88 Please reword to… At the end of the treatment period, animals from all experimental groups were sacrificed, …
Materials and Methods
4.1 – Can authors include the suppliers of the chemicals, as to where the chemicals were sourced from?
4.3.
What was the age of the animals (in weeks) and how many animals in total?
260 – I suggest that you use the acronym SD for Sprague-Dawley rats.
270 – please replace the decibels with Db as SI unit of measurement
272-274 – please reword this sentence to more accurately describe the acclimatization process. The acclimatization process is usually referred to once the animal arrive to the facility (reduction of stress, etc) rather than the acclimatization to the diet (specifically the diet that is used as a treatment).
280 – According to the statement (please see minor comments section comment about acclimatisation) it appears that the NC rats were than put on the standard rats chow (which is usually LF) but this is only after the HFHC treatment initially. If this is correct, then the NC group is not really representing the control group? Please clarify and elaborate more.
314 – Can you please check the temperature as it is stated to be 1000C, I believe this is typographical error and it might be 100Deg C.
References
Please update the reference list with newer articles as there is only one article published in 2018. In addition, I suggest that authors review the reference list for consistency.
Author Response
To whom it may concern
On behalf of all the authors, I would like to express my gratitude to the reviewer for their valuable time and contribution towards our manuscript. We appreciate the input you have given to improve our manuscript.
We have implemented all the typographical/grammatical corrections suggested by the reviewer and considered all the scientific recommendations made. All the authors have seen the changes to make sure the suggestions were addressed accordingly.
Attached is the corrected manuscript and the response to the corrections/suggestions
Sincerely
Dr A Khathi
Corresponding author

Reviewer 2 Report
The paper is well-written and potentially interesting data are presented. The paper is descriptive in nature and would benefit if mechanisms are presented. The glucose lowering effect is robust and this can be further investigated as mechanism of action of OA.
Table 1. Please include the heart weight to BW ratio.
Diet. The constituents should be included in a table, along with food intake.
Author Response

(The authors gave the same response as above.)

Reviewer 3 Report
In a pre-diabetes rat model, the Authors investigate the effects of the plant-derived oleanolic acid, in the presence/absence of dietary intervention, on selected cardiovascular risk factors, in comparison with control untreated animals. The same protocol was applied to metformin-treated pre-diabetes animals. The manuscript provides additional insights on the protective role of natural compounds, a field potentially promising that received great attention in the last years.
There are however some aspects that should be addressed by the Authors.
General Comments.
1. As specified below in details, the description of some results in the text does not correspond to the quantitative data reported in the figures. Thus, some sentences must be changed in the description of the results.
2. Different treatments have been analysed, metformin (with and without dietary intervention, Met and Met+DI) and oleanolic acid (OA and OA+DI), however: a) it should be mentioned that in some cases both treatments required DI to obtain the recovery of the parameters, and b) no discussion about the comparison between metformin and OA is provided. Differences and possible advantages of OA administration must be highlighted in the discussion.
Specific Comments.
1. Abstract. Line 12: …prediabetes is associated “with”…..(the word with is lacking)
2. Abstract line 24: “the administration of OA, in both the presence and absence of dietary intervention, showed a significant decrease… “ . As specified below, this is not always true, in some cases DI is necessary to see a positive effect. This sentence must be attenuated, accordingly.
3. Figure 1. Can the Authors provide additional measurements of blood glucose to show a time-depending behaviour of this effect?
4. Table 2 and the related sentence in the results lines: 108-109. Please check data and reported significant differences vs. PC for LDL-C, in OA and OA+DI. From the table it appears that OA+DI is much more effective than OA. In addition, no symbols denoting significant differences vs. PC is reported in the table.
Results line 131: typing error: “concertation should be “concentration”
5. Results lines 135-136: The significant increase in SOD and GPx is comparable in OA and OA+DI and Met+DI, while Met treatment alone is not effective. The text should be corrected, accordingly.
6. Discussion: check the text where the effects of OA and OA+DI are discussed, in accordance with the changes suggested in the previous points of concern.
7. Discussion: add a short discussion aimed at comparing metformin and OA treatments, as indicated in the general comments.
8. Methods lines 287 (formula): the formula is not correct.
MAP = SBP + 2 (DBP)
3
Author Response

(The authors gave the same response as above.)

Round 2
Reviewer 1 Report
Dear Authors
Congratulations on the improved manuscript.
Kind Regards
Reviewer 2 Report
The authors have addressed my comments. However, the effects of OA on glucose metabolism deserve more attention.